# The Relationship between Laissez-Faire Leadership and Cyberbullying at Work: The Role of Interpersonal Conflicts

**DOI:** 10.3390/bs14090824

**Published:** 2024-09-16

**Authors:** Alfonso Cárdenas-Miyar, Francisco J. Cantero-Sánchez, José M. León-Rubio, Alejandro Orgambídez-Ramos, Jose M. León-Pérez

**Affiliations:** 1Department of Social Psychology, Universidad de Sevilla, 41004 Sevilla, Spain; acardenas@us.es (A.C.-M.); fcantero@us.es (F.J.C.-S.); jmleon@us.es (J.M.L.-R.); 2Department of Social Psychology, Universidad de Malaga, 29071 Malaga, Spain; aorgambidez@uma.es

**Keywords:** workplace cyberbullying, cyberbullying at work, cyberincivility, gender differences, laissez faire leadership, interpersonal conflicts

## Abstract

A person can experience cyberbullying at work when exposed to repetitive and intrusive negative acts facilitated by new information and communication technologies (ICTs). The incidence of workplace cyberbullying has rapidly increased following the COVID-19 pandemic. This issue does not arise in isolation; leadership plays a critical role. Leaders who fail to set clear rules and provide minimal supervision may exacerbate interpersonal conflicts among subordinates. This study explores the role of laissez-faire leadership and interpersonal conflicts on workplace cyberbullying from a gender perspective. A two-wave panel study was conducted (*N* = 1995; 53.6% women; M age = 42.02 years old; SD = 9.23; age range: 18–74 years old). Our findings indicate no direct relationship between laissez-faire leadership and workplace cyberbullying; however, there is a significant indirect relationship. Laissez-faire leadership is associated with a higher frequency of interpersonal conflicts, which in turn are related to cyberbullying, making interpersonal conflicts a mediator. Additionally, gender moderates the relationship between interpersonal conflicts and workplace cyberbullying. Our results suggest that interpersonal conflicts may increase exposure to cyberbullying, particularly for men under laissez-faire leadership. These findings have managerial implications for introducing tailored interventions to prevent workplace cyberbullying.

## 1. Introduction

As time passes, the work environment is continually updated. This is illustrated by the introduction of Information and Communication Technology (ICT) into the workplace, a tool that has been increasingly present since the turn of the century. A significant acceleration was seen in the use of ICT in the workplace during the COVID-19 pandemic, which forced many companies and workers to switch to digital operations and systems [1]. ICT tools in the workplace have transformed behaviour patterns and work styles since their implementation, enabling higher-quality and more convenient messaging, the use of storage applications, and remote collaboration [2].

Unfortunately, the negative behaviours that already exist in the workplace have also evolved with the massive introduction of ICT tools at work [3]. This is the case of cyberbullying, which refers to the consistent occurrence of “negative acts stemming from work relationships that occur through the use of ICT tools” [4]. Cyberbullying may differ from traditional face-to-face workplace bullying and harassment due to the lack of nonverbal cues in social interaction through ICT tools like text messages and audiovisual content on mobile phones, emails, or social media [5]. In addition, cyberbullying can be more intrusive and persistent, making it difficult to disconnect and recover from work because (a) the permanent digital connection some employees experience blurry the boundaries between work and private spheres and (b) due to the inherent characteristics of the internet, once a comment or piece of content is posted on the web, it can circulate indefinitely (unless it is actively removed).

Therefore, cyberbullying is considered one of the greatest stressors that can be found in the workplace environment because the negative acts are carried out repeatedly over a period of time and, even when may be performed just once, intrude into the victim’s personal life, having the potential to expose private information over the internet to an online audience [4,6,7,8]. Previous studies have shown that being exposed to cyberbullying at work is related to health impairment (e.g., increase in stress, anxiety, and exhaustion; problems with sleep quality such as insomnia; or psychosomatic symptoms such as headaches) and a decrease in job satisfaction, work engagement, and creativity [9,10,11,12,13].

Furthermore, recent studies have highlighted that cyberbullying at work affects a great number of employees, showing high prevalence rates. For example, Kowalski et al. [14] reported that 20% of their study participants (out of a total of 3699 individuals) identified themselves as victims of workplace cyberbullying. Similarly, Park et al. [15] indicated that 34% of their sample received between one and three emails containing cyberbullying content daily. Vranjes et al. [4] found that 8 out of 10 people had experienced workplace cyberbullying in the last 6 months, and between 14% and 20% felt they had been victims in their work environment in the past weeks.

Considering cyberbullying high prevalence and its detrimental consequences on employee well-being and, consequently, organisational productivity, it is surprising the lack of evidence on the process of cyberbullying at work [16,17]. A better knowledge of workplace cyberbullying antecedents and processes is crucial to effectively prevent and manage it.

For that reason, given that cyberbullying has several similarities with traditional workplace bullying, our goal is to explore whether the previous well-established “work environment hypothesis”, which suggests that factors in the work environment enable and trigger bullying behaviours, may be applied to cyberbullying research. Furthermore, we propose that laissez-faire leadership facilitates the emergence and escalation of interpersonal conflicts into cyberbullying behaviours at work.

### 1.1. Laissez-Faire Leadership and Cyberbullying

According to the Multifactor Leadership model, leadership can be understood through three distinct dimensions: transformational, transactional, and laissez-faire leadership (see [18]). While transformational leadership inspires enthusiasm, innovation, and loyalty through charisma, intellectual stimulation, and individualised consideration, and transactional leadership relies on contingent rewards and corrective feedback to maintain performance, laissez-faire leadership is marked by the absence of leadership itself. As described by Bass and Avolio [19], laissez-faire leadership is characterised by passive behaviours, a lack of intervention, delayed decision-making, and an overall failure to provide feedback or motivation to subordinates.

The first conceptualisations of workplace bullying considered the fact that bullying emerges in organisations with work design deficiencies and negative leadership and social climate [20,21,22]. Indeed, when Einarsen et al. [23] validated the workplace bullying questionnaire NAQ-R in a sample of 5288 UK employees, they included measures of autocratic and laissez-faire leadership to test the construct validity of the NAQ-R. They found significant correlations between laissez-faire leadership and both person-related and work-related bullying behaviours. In addition, Hoel et al. [24] reported a positive correlation between laissez-faire leadership and self-reported bullying among British workers.

Similarly, Hauge et al. [25] confirmed the ‘work environment hypothesis’ of workplace bullying as they found that role conflict, interpersonal conflicts, and negative leadership (including laissez-faire leadership) were the strongest predictors in a cross-sectional study based on a representative sample of the Norwegian workforce (*N* = 2539). Furthermore, Skogstad et al. [26] also reported a direct effect of laissez-faire leadership and workplace bullying. They pointed out that laissez-faire leadership behaviours, such as the absence of feedback, rewards, and involvement, may be understood by subordinates as systematic neglect and ignorance. Therefore, laissez-faire leadership has been recognised as a destructive leadership style associated with a higher risk of workplace bullying (for a meta-analysis see [27]).

In line with previous studies, we extend the ‘work environment hypothesis’ to the context of workplace cyberbullying. Under the premise that laissez-faire leadership is characterised by the absence of oversight, it may allow harmful online behaviours to be established in the workplace. Thus, we hypothesise that the presence of a laissez-faire leader will result in an increase in exposure to workplace cyberbullying behaviours.

**H1.** *Laissez-faire leadership is positively associated to exposure to cyberbullying behaviours at work*.

### 1.2. Interpersonal Conflicts and Cyberbullying

When addressing workplace cyberbullying, it is essential to consider interpersonal conflict as a key element. Leymann [22] identified frustrating working conditions and poorly managed interpersonal conflicts as the primary antecedents of workplace bullying. From a conflict perspective, workplace bullying can be understood as a prolonged, intractable, escalating, destructive conflict between individuals with a power differential ([28]). These types of conflicts are among the most difficult to manage and resolve, as they are deeply rooted in the organisation’s policies, practices, procedures, work structure, and labour relations ([29]).

The ‘conflict escalation model’ of workplace bullying provides a useful framework for understanding how conflicts can escalate into deviant behaviours such as cyberbullying (see [30]). According to this model, when task-related disagreements or factual disputes about how specific tasks should be accomplished are managed unsuccessfully, the focus shifts to personal differences between the individuals involved, which is characterised by tension, frustration, and animosity among conflict parties. Again, if interpersonal conflicts are not successfully managed at this stage, conflicts escalate to more destructive levels, including bullying.

Empirical studies have differentiated interpersonal conflicts and workplace bullying and have supported such conflict escalation model: relationship conflict mediates the effect of task-related conflict on workplace bullying [31,32,33]. In a more recent intensive longitudinal study, which collected data from 57 military naval cadets over 30 consecutive days, revealed that experiencing interpersonal conflicts is associated with exposure to bullying behaviours during the same day [34].

Based on previous research findings, we consider that the conflict escalation model also applies to other negative acts like cyberbullying behaviours, and therefore, we propose that interpersonal conflicts will result in an increase in exposure to workplace cyberbullying behaviours.

**H2.** *Interpersonal conflicts are positively associated with exposure to cyberbullying behaviours at work*.

In addition, as cyberbullying is a social phenomenon, it is essential to examine sex and gender differences. In that sense, a recent meta-analysis examined the sex differences in perceived workplace mistreatment [35]. Interestingly, findings revealed that, although men and women experience similar levels of bullying, men perceive significantly more interpersonal conflicts at work than women. Similarly, recent studies on cyberbullying in the workplace further reveal sex and organisational differences that diverge from face-to-face bullying research. For example, previous studies found that men and managers were more frequently exposed to cyberbullying than women and non-managers [36,37]. Therefore, we propose that sex will moderate the relationship between interpersonal conflicts and cyberbullying at work.

**H3.** *Sex moderates the relationship between interpersonal conflicts and workplace cyberbullying, with male employees experiencing higher levels of cyberbullying than female employees*.

### 1.3. Laissez-Faire Leadership, Interpersonal Conflicts and Cyberbullying

Despite the direct effects of laissez-faire leadership on workplace bullying, there is empirical evidence from qualitative research that suggests indirect effects through interpersonal conflicts (i.e., conflict escalation process, [30,31], particularly in digital contexts where social cues are reduced [38]. In this sense, destructive leadership may create a stressful and frustrating environment in which interpersonal tensions and conflicts emerge. For example, findings from a two-year prospective study indicate that laissez-faire leadership “increases the rate by which bullying targets continue to be victimized over time” (p. 304) [39]. Also, a recent daily diary study by Agotnes and colleagues [40] found that laissez-faire leadership amplified the relationship between experiencing work pressure and exposure to negative acts at work. Furthermore, high levels of laissez-faire leadership have also been linked to lower rates of managerial intervention in bullying cases. Since laissez-faire leadership is characterised by conflict-avoidant leaders and a lack of clear norms, their lack of intervention in bullying cases not only fails to stop but even exacerbates this detrimental situation [41].

In sum, previous studies have consistently shown that laissez-faire leadership is associated with higher levels of interpersonal conflicts, fostering a work environment that provides fertile ground for workplace bullying. Similarly, we propose that laissez-faire leadership may facilitate the emergence and escalation of interpersonal conflicts into cyberbullying behaviours.

**H4.** *Laissez-faire leadership is positively associated with interpersonal conflicts at work*.

**H5.** *Interpersonal conflicts mediate the relationship between laissez-faire leadership and exposure to workplace cyberbullying behaviours*.

### 1.4. Aims and Goals

Recent mistreatment and cyberbullying literature emphasise the need to analyse contextual factors, investigating the dynamics between different layers of the organisation as well as other factors, such as sex differences, that may differ from face-to-face workplace bullying [8,42].

In response, and considering that the relationship between laissez-faire leadership, interpersonal conflicts, and workplace bullying over time has not yet been thoroughly examined in the context of workplace cyberbullying, we conducted a two-wave panel study with a three-month time lag between measures, aiming to test whether interpersonal conflicts mediate the relationship between laissez-faire leadership and workplace cyberbullying (see Figure 1).

Our findings may offer valuable insights into the dynamics of workplace cyberbullying, providing occupational health and human resources professionals with relevant information for implementing more accurate preventive measures and effectively managing cyberbullying at work.

## 2. Materials and Methods

### 2.1. Design and Participants

We conducted a two-wave panel study in a general working Spanish population. The data collection method was online, allowing us to reach a larger sample size and, consequently, to test our hypotheses in a more representative population. This approach enabled a more robust analysis and more generalisable results. The two-wave panel study, with a three-month interval between waves, initially comprised a sample size of 5000 participants. From such participants, 1995 completed the same questionnaire three months later (response rate of 39.9%; 53.6% women; M age = 42.02 years old; SD = 9.23; age range: 18–74 years old).

Regarding the characteristics of our sample, 13.0% of our participants were self-employed (vs. 87.0% employed). Most of the participants reported having a permanent contract (84.2% vs. 14.8% temporary contract and 1.0% training and apprenticeship contract) and having a job experience higher than 5 years (56.6%). Additionally, 80.8% of the participants reported having regular contact with clients or users (vs. 19.2% reported not having regular contact). In terms of remote work, 61.6% did not work from home (*n =* 1229), 19.5% worked from home one or two days a week (*n =* 390), 9.2% worked from home three or four days a week (*n =* 183), and 9.7% worked exclusively from home (*n =* 193).

When considering job level relative to education, 10.5% reported that their job was above their educational level (*n =* 210), 27.9% reported that their job was below their educational level (*n =* 557), and 61.6% reported that their job corresponded to their educational level (*n =* 1228). Regarding promotions, 42.7% had been promoted since joining the company (*n =* 852), while 57.3% had not been promoted (*n =* 1143).

Finally, company size distribution was as follows: 17.7% worked in self-employed or companies with fewer than 10 employees (*n =* 353), 20.7% in companies with 10 to 50 employees (*n =* 413), 22.8% in companies with 50 to 250 employees (*n =* 455), and 38.8% in companies with more than 250 employees (*n =* 774).

### 2.2. Measures

Laissez-faire leadership was measured using seven items designed to capture this leadership style (see [43]). Sample items included “My supervisor is often absent when needed” and “My supervisor has a strong tendency to postpone decision-making”. Respondents were asked to indicate their level of agreement with each statement on a 5-point Likert scale, where 1 = Strongly disagree, 2 = Disagree, 3 = Neither agree nor disagree, 4 = Agree, and 5 = Strongly agree. Overall, the mean scores for the scale reflected the extent to which supervisors exhibited laissez-faire leadership behaviours. The internal consistency was acceptable, as Cronbach’s alpha was 0.88.

Interpersonal conflicts were measured using six items from the Interpersonal Conflict Scale (ICS-14) [44], designed to capture the nature and frequency of such conflicts. Sample items included “My colleagues and I differ in our views on the cause and solution of work-related problems”, “There are personal frictions between my colleagues and me” and “My colleagues and I disagree on opinions about the work being done”. Respondents were asked to indicate how often these conflicts occurred on a 4-point Likert scale, where 1 = Never, 2 = Sometimes, 3 = Often, and 4 = Always. Overall, the mean scores for the scale reflected the prevalence and intensity of interpersonal conflicts within the workplace. The Cronbach’s alpha was 0.82.

Cyberbullying at work was measured using six items from the Inventory of Cyberbullying Acts at Work (ICA-W) [45]. Sample items included “Someone hinders your work by withholding information from emails or computer files that you need and does not share with you”, “Rumours and gossip about you have been spread through new technologies (internet, email, social networks, instant messaging groups like Signal or WhatsApp)” and “Personal or private information about your life has been distributed or shared without your consent through new technologies (internet, email, social networks, instant messaging groups like Signal 6.2. or WhatsApp 15.2)”. Respondents were asked to indicate how often they had experienced each of these behaviours on a 5-point Likert scale, where 1 = Never, 2 = Occasionally, 3 = Monthly, 4 = Weekly, and 5 = Daily. Overall, the mean scores for the scale reflected the frequency and severity of cyberbullying behaviours encountered by employees. The internal consistency was 0.94.

### 2.3. Data Analysis

First, we conducted some preliminary and descriptive statistics, Then, we conducted a moderated-mediation regression analysis using the PROCESS macro, model 14 in SPSS v. 26 [46]. With the aim of both replicating and assuring the validity of the process, the analyses were also conducted in R by using the software Jamovi 2.3.28 (outcomes can be obtained from the first author). A moderated-mediation analysis is particularly suitable for this study for several reasons: (a) this method enables us to investigate not just direct relationships but also complex relations such as the interactions between different variables; (b) by considering both the mediating role of interpersonal conflicts and the moderating role of sex/gender, we can gain a deeper understanding of how these factors interact to influence workplace cyberbullying; (c) moderated-mediation allows us to specifically test the pathway from laissez-faire leadership to cyberbullying through interpersonal conflicts, and how this pathway is affected by sex/gender. This specificity is essential for developing targeted interventions; and (d) by using this method, we can separate the direct effects from the indirect effects and understand the conditional nature of these effects. This is particularly important in organisational research where multiple variables often interact in complex ways.

Thus, we introduced laissez-faire leadership at time 1 as the predictor variable, interpersonal conflicts at time 2 as the mediator variable; cyberbullying at work at time 2 as the dependent variable, and sex/gender (dichotomised: 0 = women, 1 = men) as a moderator variable on the conflicts-bullying path.

## 3. Results

First, we conducted some descriptive analyses. As can be seen in Table 1, there is a significant positive correlation between all variables and workplace cyberbullying. Also, there is a significant positive correlation between laissez-faire leadership and interpersonal conflicts.

Then, we conducted a moderated mediation regression analysis to test our hypotheses. The results regarding the direct effect of laissez-faire leadership on workplace cyberbullying indicated that there is no significant correlation between both variables (β = 0.0234, *p* = 0.1299; 95%CI = −0.0069; 0.0536). This implies that laissez-faire leadership alone does not directly predict the occurrence of workplace cyberbullying, and therefore data do not support our hypothesis 1 (H1).

In contrast, interpersonal conflicts are significantly associated to workplace cyberbullying (β = 0.6253, *p* < 0.001; 95%CI = 0.5573; 0.6933), which supports our second hypothesis (H2). Furthermore, there is an interaction effect between interpersonal conflicts and sex on workplace cyberbullying (β = 0.2515, *p* < 0.001; 95%CI = 0.1527; 0.3504), which also supports our third hypothesis (H3). As can be seen in Figure 2, the effect of interpersonal conflicts on workplace cyberbullying was more pronounced for men (β = 0.8768; 95%CI = 0.8027; 0.9510; *p* < 0.001) than for women (β = 0.6253; 95%CI = 0.5573; 0.6933; *p* < 0.001).

Finally, our results demonstrated that laissez-faire leadership had a significant positive correlation with interpersonal conflicts (β = 0.1548, *p* < 0.001; 95%CI = 0.1297; 0.1800), which supports our fourth hypothesis (H4). Furthermore, our results indicate that interpersonal conflicts represent a pathway through which laissez-faire leadership influence workplace cyberbullying, which supports hypothesis 5 (H5). This indirect effect of laissez-faire leadership on workplace cyberbullying through interpersonal conflicts was more pronounced for men (β = 0.1358; 95%CI = 0.1050; 0.1691) than for women (β = 0.0968; 95%CI = 0.0724; 0.1235). These findings indicate that men are more likely than women to experience increased workplace cyberbullying as a result of interpersonal conflicts stemming from laissez-faire leadership. Overall, our model predicts around 32% of the variance of workplace bullying (R^2^ = 0.32; F(4, 1990) = 234.55; *p* < 0.001). Figure 3 below offers a summary of the model.

## 4. Discussion

This study focuses on the analysis of the phenomenon of workplace cyberbullying, specifically aiming to contribute to the scientific literature by exploring potential mechanisms that can explain how cyberbullying emerges in organizational contexts [17].

Our findings suggest that while laissez-faire leadership does not directly predict workplace cyberbullying, it significantly contributes to experiencing interpersonal conflicts, which, in turn, leads to higher exposure to cyberbullying behaviours. Hence, although laissez-faire leadership has been characterised in the literature as a precursor of workplace bullying [47], our findings suggest that this type of leadership has a more indirect relationship with cyberbullying at work.

A potential explanation is that cyberbullying is a process that requires longitudinal designs to be observed. Indeed, while previous cross-sectional studies have included laissez-faire leadership as a predictor of workplace bullying [24,26,48,49], longitudinal studies have often failed to replicate such relationships and have included laissez-faire leadership as a moderator or “condition under which the bullying process can endure and progress” (p. 297) [39]. Following this rationale, our findings from a two-wave panel design with three months between measures point out that interpersonal conflicts that emerge in groups under laissez-faire leadership can also be a potential mechanism that explains cyberbullying at work. In other words, as expected, being under a leader who avoids making decisions, resists expressing opinions, hesitates about taking action, and is absent when needed [19] fosters negative interactions and a hostile climate characterised by high levels of interpersonal conflict [39,50]. These unresolved interpersonal issues, when not adequately addressed or mediated by the leader, are likely to persist over time and end up in a bullying situation [30,31,51]. Indeed, these unmanaged conflicts are more likely to escalate, particularly when there is an absence of effective leadership intervention [40]. This assumption was also observed by Salin and colleagues [52], who concluded that, in working environments where competition is promoted, laissez-faire leadership facilitates incivility behaviours among workers. Indeed, in their study, they stated that “high-performance work practices were associated with more incivility through competition, but only under conditions of high laissez-faire leadership” (p. 6). Future studies should explore contextual factors that may explain the boundary conditions in which cyberbullying at work emerges.

At the individual level, our findings have implications for the Conservation of Resources [53]. It is not laissez-faire leadership itself that directly causes workplace cyberbullying, but rather the continuous depletion of resources that makes other factors more likely to emerge and become problematic. This resource depletion occurs because employees are unable to recover or maintain their resources in a laissez-faire leadership environment. The persistent lack of guidance and support from leaders forces employees to expend their personal resources, leading to increased stress and conflict.

In addition, our findings revealed that the impact of these interpersonal conflicts on cyberbullying is more pronounced among men. In the literature on workplace cyberbullying, gender differences have been noted, with different outcomes observed based on the type of uncivil actions occurring at work [37,54,55,56]. Moreover, Forsell [36] found that gender differences are more pronounced in workplace cyberbullying than in traditional workplace harassment. In our study, the relationship between interpersonal conflicts and cyberbullying is enhanced in the case of men compared to women. This result aligns with a previous meta-analysis of sex differences in perceived mistreatment at work that highlights that men report higher levels of interpersonal conflicts than women, whereas women experience more harassment and discrimination at work [35]. In a similar vein, regarding studies focusing on the categorisation of various acts of cyber incivility, D’Souza et al. [57] demonstrated that men have a higher prevalence of experiencing workplace cyberbullying, whereas women are more likely to experience other negative behaviours, such as cyber harassment and cyber discrimination.

Additionally, Weber et al. [58], in a scenario-based experiment, found that female victims of workplace cyberbullying were more likely to receive help and were less likely to be blamed than male victims, particularly if the perpetrator was a man. From a sociocultural perspective, they explained their findings based on the existing sexist attitudes in some societies, which leads participants to be more empathetic toward a female victim of workplace cyberbullying. Consequently, future research should explore if male victims of cyberbullying might be marginalised by their social environment. In doing so, future research should consider using multicultural populations or conducting cross-cultural comparisons to gain a more holistic view of the phenomenon.

At the group level, these findings underscore the importance of targeted interventions aimed at managerial positions and leaders of various work groups, not only to reduce workplace cyberbullying but to prevent it and promote a healthy work environment in which a psychological safety climate exists (see [59,60]. Indeed, previous studies have shown that leaders who are supportive and promote a psychological safety climate, in which workers share the perception that their leaders care about and protect their well-being, retain workers employed in the opposite sex-dominated occupations (e.g., [61]). Indeed, emphasising the need for a work environment based on equality is crucial for protecting employees’ psychological health. In that sense, our findings suggest that both conflict management and leadership interventions seem crucial to creating a psychologically safety climate and preventing workplace cyberbullying [62,63].

Future studies should continue exploring workplace cyberbullying as a group phenomenon and overcome the limitations of our design. For example, although our design implies two waves of data collection in a representative working population sample, future research could benefit from using a greater number of measurement points to better infer causality and test potential reverse causal effects. Additionally, while this study focuses on self-reported data from employees, future studies may include information from leaders and team members, checking for multilevel associations to better capture group dynamics over time [64].

Finally, at the organisational level, our findings highlight the importance of carefully designing career development plans and capacitation training programs that focus on social skills, including conflict management skills [65,66]. Also, future research may benefit from incorporating organisational level variables, such as organisational culture, and contextualising sex/gender effects depending on the organisational setting (i.e., male- vs. female-dominated occupations) in which cyberbullying occurs [67,68]. Moreover, it might be interesting to include different forms of mistreatment at work (i.e., online and offline bullying) in the future to capture a more complete picture of the situation.

## 5. Conclusions

Having discussed the various aspects of the research, the importance of this article as a contribution to understanding the emerging phenomenon of workplace cyberbullying across work sectors is evident. Our findings emphasise the necessity of considering both direct and indirect pathways, as well as the moderating role of sex/gender, in order to gain a comprehensive understanding of the dynamics of workplace cyberbullying. Indeed, this study highlights important relationships between leadership style, the emergence of interpersonal conflicts, and the potential entrenchment and escalation of these conflicts, making workplace cyberbullying more likely. Additionally, it provides a gender perspective, offering a more precise picture of the current state of the phenomenon.

## Figures and Tables

**Figure 1 behavsci-14-00824-f001:**
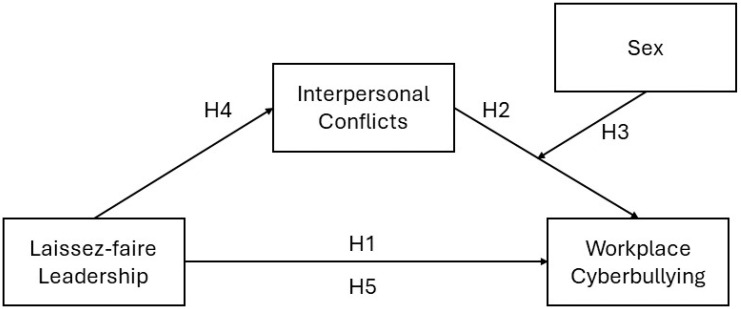
Conceptual model.

**Figure 2 behavsci-14-00824-f002:**
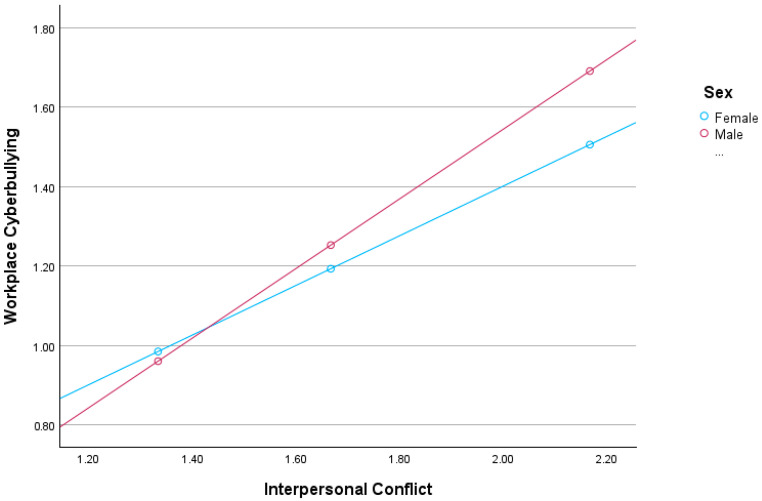
Moderation of sex on the mediated relationship between laissez-faire leadership and workplace bullying through interpersonal conflicts.

**Figure 3 behavsci-14-00824-f003:**
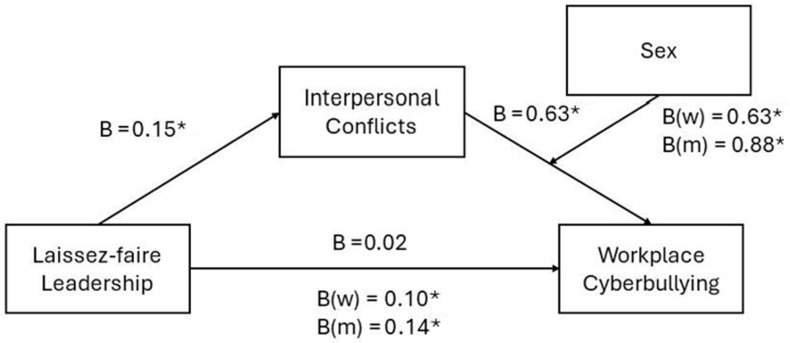
Beta coefficients of the model. Note: * *p* < 0.01.

**Table 1 behavsci-14-00824-t001:** Means, standard deviations, and bivariate correlations between study variables (*N =* 1995).

Variable	M	SD	1	2	3	4
1. Sex	0.46	0.50	-	−0.01	0.02	0.08 *
2. Laissez faire leadership	2.65	0.86		-	0.26 *	0.17 *
3. Interpersonal conflict	1.79	0.51			-	0.55 *
4. Cyberbullying	1.32	0.69				-

Note: * *p* < 0.01.

## Data Availability

Following open access and FAIR data principles, raw data can be found at: https://osf.io/5j48b/ accessed on 13 May 2024.

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
