# Peer review of "The Relationship between Laissez-Faire Leadership and Cyberbullying at Work: The Role of Interpersonal Conflicts"

_behavsci, 2024, doi:10.3390/bs14090824_

Round 1

Reviewer 1 Report

Comments and Suggestions for Authors

Dear Authors

I found this paper an enjoyable read. It is very clear and concise.  The mix of pre COVID and post COVID literature positioned the issues well.   Previous studies were documented, and a gap established. Definitions of key concepts were presented throughout, methods were clearly explained, and demographics presented in detail which is important for this design. Analysis was well documented.

Main findings and other findings were logically presented.     

I found the gendered discussion interesting, and a good set of future research directions presented. I would recommend authors to look at the "psychological safety" literature to add more depth to this discussion.

The visuals presented were not very sophisticated and I would recommend the authors look to providing more analytical rigor to these (Figures 1 & 2).

I hope these comments assist the authors to improve the manuscript.                                           

Comments on the Quality of English Language

Minor editing

Author Response

Dear Reviewer 1,

Thank you for your time and consideration. We really appreciate your suggestions to improve our manuscript. Please find a detailed response to all your comments below:

Comment 1: I found this paper an enjoyable read. It is very clear and concise.  The mix of pre COVID and post COVID literature positioned the issues well.  Previous studies were documented, and a gap established. Definitions of key concepts were presented throughout, methods were clearly explained, and demographics presented in detail which is important for this design. Analysis was well documented. Main findings and other findings were logically presented.     

Response 1: Thank you for your feedback and positive comments.

Comment 2: I found the gendered discussion interesting, and a good set of future research directions presented. I would recommend authors to look at the "psychological safety" literature to add more depth to this discussion.

Response 1: Thank you for your suggestion. We have opted to include the psychological safety climate literature in the conclusions: “These findings underscore the importance of targeted interventions aimed at managerial positions and leaders of various work groups, not only to reduce workplace cyberbullying but to prevent it and promote healthy work environment in which a psychological safety climate exists (see Mullen et al., 2004; Plimmer et al., 2022). In-deed, previous studies have shown that leaders who are supportive and promote a psychological safety climate, in which workers shared the perception that their leaders care about and protect their well-being, retain workers employed in the opposite sex-dominated occupations (e.g., Halliday et al., 2022). Indeed, emphasizing the need for a work environment based on equality is crucial for protecting employees' psycho-logical health. In that sense, our findings suggest that both conflict management and leadership interventions seem crucial to create a psychological safety climate and pre-vent workplace cyberbullying (e.g., Berglund et al., 2024; Leon-Perez et al., 2012, 2016).” (p. 10).

Comment 3: The visuals presented were not very sophisticated and I would recommend the authors look to providing more analytical rigor to these (Figures 1 & 2).

Response 3: We agree with your comment. Thus, we have improved our Figures (and added a new one to summarize our results; see p. 7 and 8).

Reviewer 2 Report

Comments and Suggestions for Authors

The article proposed for evaluation: The relationship between laissez-faire leadership and cyber-2 bullying at work: The role of interpersonal conflicts is verry intersting, but are many weak points which msut be improved in order to be proposed for publication.

1.In the final part of the Introduction, the authors need to add a final part which is consisting in making a short presentation of each analysed section.

2.Between Introduction and Materials and Methods, the authors need to add another section, called Literature review, research hypothesis and conceptual model. In this section the authors must add the sub-section according to the number of hypothesis, and these hypothesis need to be added on the model.

After each sub-section, the literature review need to be written, adding new sources, and resesrch in the field, based on the analysed influence. At the end of each sub-section must be established an hypothesis, which must be added in Conclusion if each hypothesis was or not accepted, according to calculations, in the end of this new section, must move the conceptual model from Data analysis (p.5) to section 2.

3.At sub-section 2.1. the authors need to add a Table in which are putted information about the respondents, called Demographic characteristics of respondents.

4.At sub-section 2.2. the authors need to add a Table in which are developed the affirmations on each analysed variables proposed in the model.

5.At section 3 after Table 2 must be introduced another Table in which must be calculated the values specific to demostrae that each proposed hypothesis is accepted or not, and the mediation if is accepted or not.

6.Discussion section must be improved, the authrs need to add Implications for each involved part in the study- for leaders, for employees, for managers, for organizations, or society. An dalso developed future research directions.

7.The sources must be added and updated, having in view: that section 2 Literature review and hypotehesis developmemnt, and that are old- only two sources are from 2024.

Having these proposals in view, the article needs major improvements and reconsider after major revisions.

Author Response

Dear Reviewer 2,

Thank you for your time and consideration. We really appreciate your suggestions to improve our manuscript. Please find a detailed response to all your comments below:

Comment 1: In the final part of the Introduction, the authors need to add a final part which is consisting in making a short presentation of each analysed section.

Response 1: Thank you for your comment. We have restructured the paper following your suggestions.

Comment 2:.Between Introduction and Materials and Methods, the authors need to add another section, called Literature review, research hypothesis and conceptual model. In this section the authors must add the sub-section according to the number of hypothesis, and these hypothesis need to be added on the model.

After each sub-section, the literature review need to be written, adding new sources, and resesrch in the field, based on the analysed influence. At the end of each sub-section must be established an hypothesis, which must be added in Conclusion if each hypothesis was or not accepted, according to calculations, in the end of this new section, must move the conceptual model from Data analysis (p.5) to section 2.

Response 2: We agree that the introduction was a bit short. Therefore, following your suggestions we have added some subsections in which we review the literature and formulate our hypotheses (see pages 2 to 4). In addition, we have included a Figure at the end of the introduction in which we summarize our hypotheses. We hope that you find the new introduction section has improved the manuscript quality.

Comment 3:At sub-section 2.1. the authors need to add a Table in which are putted information about the respondents, called Demographic characteristics of respondents.

Response 3: Thank you for your suggestion. However, we have opted for keeping the demographic characteristics reported in the text as a Table would not add value. We consider that we have detailed the demographic characteristics of our sample according to APA format guidelines (see page 5).

Comment 4:At sub-section 2.2. the authors need to add a Table in which are developed the affirmations on each analysed variables proposed in the model.

Response 4: Thank you for your suggestion. According to your suggestion we have included a Table on page 7.

Comment 5: At section 3 after Table 2 must be introduced another Table in which must be calculated the values specific to demostrae that each proposed hypothesis is accepted or not, and the mediation if is accepted or not.

Response 5: Thank you for your comment. However, we consider that it is enough to describe the statistics of our analysis in the text. Anyway, we have revised the results section and we have added a Figure to summarize the results of our model (see page 8).

Comment 6:Discussion section must be improved, the authrs need to add Implications for each involved part in the study- for leaders, for employees, for managers, for organizations, or society. An dalso developed future research directions.

Response 6: Following your suggestion, we have improved our discussion to clearly identify the implications of our findings (For example: “These findings underscore the importance of targeted interventions aimed at managerial positions and leaders of various work groups, not only to reduce workplace cyberbullying but to prevent it and promote healthy work environment in which a psychological safety climate exists (see Mullen et al., 2004; Plimmer et al., 2022). In-deed, previous studies have shown that leaders who are supportive and promote a psychological safety climate, in which workers shared the perception that their leaders care about and protect their well-being, retain workers employed in the opposite sex-dominated occupations (e.g., Halliday et al., 2022). Indeed, emphasizing the need for a work environment based on equality is crucial for protecting employees' psycho-logical health. In that sense, our findings suggest that both conflict management and leadership interventions seem crucial to create a psychological safety climate and pre-vent workplace cyberbullying (e.g., Berglund et al., 2024; Leon-Perez et al., 2012, 2016).”, see page 9). In addition, future research directions have been highlighted. For example: “This assumption was also observed by Salin and colleagues (2022), who concluded that, in working environments where competition is promoted, a laissez-faire leader-ship facilitates incivility behaviours among workers. Indeed, in their study “high-performance work practices were associated with more incivility through com-petition, but only under conditions of high laissez-faire leadership” (p. 6). Future stud-ies should explore contextual factors that may explain the boundary conditions in which cyberbullying at work emerges.” (p. 8). Another example is “Indeed, it seems important to contextualize sex/gender effects depending on the or-ganizational setting (i.e., male- vs. female-dominated occupations) in which cyberbul-lying occurs (Forsell, 2018; Salin & Hoel, 2013).” (see p. 8).

Comment 7:The sources must be added and updated, having in view: that section 2 Literature review and hypotehesis developmemnt, and that are old- only two sources are from 2024.

Response 7: Thank you for raising this issue. Following your suggestion the introduction has been improved and updated references have been added, such as:

Jagsi, R., Griffith, K., Krenz, C., Jones, R. D., Cutter, C., Feldman, E. L., Jacobson, C., Kerr, E., Paradis, K. C., Singer, K., Spector, N. D., Stewart, A. J., Telem, D., Ubel, P. A., & Settles, I. (2023). Workplace Harassment, Cyber Incivility, and Climate in Academic Medicine. JAMA, 329(21), 1848–1858. https://doi.org/10.1001/jama.2023.7232

Leung, A. N. M., Ho, H. C., Hou, W. K., Poon, K. T., Kwan, J. L., & Chan, Y. C. (2024). A 1‐year longitudinal study on experiencing workplace cyberbullying, affective well‐being and work engagement of teachers: The mediating effect of cognitive reappraisal. Applied Psychology: Health and Well‐Being.

Mullen, J., Thibault, T., & Kelloway, E. K. (2024). Occupational health and safety leadership. In L. E. Tetrick, G. G. Fisher, M. T. Ford, & J. C. Quick (Eds.), Handbook of occupational health psychology (3rd ed., pp. 501–516). American Psychological Association. https://doi.org/10.1037/0000331-025

Mushtaq, W., Qammar, A., Shafique, I. and Anjum, Z.-U. (2022), "Effect of cyberbullying on employee creativity: examining the roles of family social support and job burnout", Foresight, Vol. 24 No. 5, pp. 596-609. https://doi.org/10.1108/FS-01-2021-0018

Platts, J., Coyne, I. and Farley, S. (2023), "Cyberbullying at work: an extension of traditional bullying or a new threat?", International Journal of Workplace Health Management, Vol. 16 No. 2/3, pp. 173-187. https://doi.org/10.1108/IJWHM-07-2022-0106

Plimmer, G., Nguyen, D., Teo, S., & Tuckey, M. R. (2022). Workplace bullying as an organisational issue: Aligning climate and leadership. Work & Stress, 36(2), 202-227.

Salin, D., Baillien, E., & Notelaers, G. (2022). High-performance work practices and interpersonal relationships: laissez-faire leadership as a risk factor. Frontiers in Psychology13, 854118.

Vranjes, I., Griep, Y., Fortin, M., & Notelaers, G. (2023). Dynamic and multi-party approaches to interpersonal workplace mistreatment research. Group & Organization Management48(4), 995-1013.

We hope that the improved version of our manuscript has covered all your concerns. Thank you again for your comments and suggestions.

Round 2

Reviewer 2 Report

Comments and Suggestions for Authors

The improvements were made by the authors, except the one refering to add Implications- theoretical and practical- for employees, for managers, for organizations (written in italic). Is important, according to the literature added by the authors, to develop these important implications, as experience in the field of the authors.

So, minor revisions are requested.

Author Response

Thank you for your time and consideration. We really appreciate that you think we were very responsive to your comments and suggestions. As you pointed out there was still one issue to be addressed. Please, find our response below.

Comment 1: The improvements were made by the authors, except the one refering to add Implications- theoretical and practical- for employees, for managers, for organizations (written in italic). Is important, according to the literature added by the authors, to develop these important implications, as experience in the field of the authors.

Response 1: Thank you for your suggestion. We have reorganized the discussion and added implications at individual, group and organizational levels (see pages 8-10): At the individual level, our findings have implications for the Conservation of Resources (Hobfoll et al., 2018). It is not laissez-faire leadership itself that directly causes workplace cyberbullying, but rather the continuous depletion of resources that makes other factors more likely to emerge and become problematic. This resource depletion occurs because employees are unable to recover or maintain their resources in a laissez-faire leadership environment. The persistent lack of guidance and support from leaders forces employees to expend their personal resources, leading to increased stress and conflict.

In addition, our findings revealed that the impact of these interpersonal conflicts on cyberbullying is more pronounced among men. In the literature on workplace cyberbullying, gender differences have been noted, with different outcomes observed based on the type of uncivil actions occurring at work (Cassidy et al., 2015; DeSouza, 2011; Gardner et al., 2016; Jagsi et al., 2023). Moreover, Forsell (2016) found that gender differences are more pronounced in workplace cyberbullying than in traditional workplace harassment. In our study, the relationship between interpersonal conflicts and cyberbullying is enhanced in the case of men compared to women. This result aligns with a previous meta-analysis of sex differences in perceived mistreatment at work that highlights that men report higher levels of interpersonal conflicts than women, whereas women experience more harassment and discrimination at work (McCord et al., 2018). In a similar vein, regarding studies focusing on the categorization of various acts of cyber incivility, D'Souza et al. (2021) demonstrated that men have a higher prevalence of experiencing workplace cyberbullying, whereas women are more likely to experience other negative behaviours such as cyber harassment and cyber discrimination.

Additionally, Weber et al. (2018), in a scenario-based experiment, found that female victims of workplace cyberbullying were more likely to receive help and were less likely to be blamed than male victims, particularly if the perpetrator was a man. From a sociocultural perspective, they explained their findings based on the existing sexist attitudes in some societies, which leads participants to be more empathetic toward a female victim of workplace cyberbullying. Consequently, future research should explore if male victims of cyberbullying might be marginalized by their social environment. In doing so, future research should consider using multicultural populations or conducting cross-cultural comparisons to gain a more holistic view of the phenomenon.

At the group level, these findings underscore the importance of targeted interventions aimed at managerial positions and leaders of various work groups, not only to reduce workplace cyberbullying but to prevent it and promote healthy work environment in which a psychological safety climate exists (see Mullen et al., 2004; Plimmer et al., 2022). Indeed, previous studies have shown that leaders who are supportive and promote a psychological safety climate, in which workers shared the perception that their leaders care about and protect their well-being, retain workers employed in the opposite sex-dominated occupations (e.g., Halliday et al., 2022). Indeed, emphasizing the need for a work environment based on equality is crucial for protecting employees' psychological health. In that sense, our findings suggest that both conflict management and leadership interventions seem crucial to create a psychological safety climate and prevent workplace cyberbullying (e.g., Berglund et al., 2024; Leon-Perez et al., 2012).

Future studies should continue exploring workplace cyberbullying as a group phenomenon and overcome the limitations of our design. For example, although our design implies two waves of data collection in a representative working population sample, future research could benefit from using a greater number of measurement points to better infer causality and test potential reverse causal effects. Additionally, while this study focuses on self-reported data from employees, future studies may include information from leaders and team members, checking for multilevel associations to better capture group dynamics over time (Leon-Perez et al., 2021).

Finally, at the organizational level, our findings highlight the importance of carefully design career development plans and capacitation training programs that focus on social skills, including conflict management skills (Cantero-Sánchez et al., 2021; León-Pérez et al., 2016). Also, future research may benefit from incorporating organizational level variables, such as organizational culture, and contextualizing sex/gender effects depending on the organizational setting (i.e., male- vs. female-dominated occupations) in which cyberbullying occurs (Forsell, 2018; Salin & Hoel, 2013). Moreover, it might be interesting to include different forms of mistreatment at work (i.e., online and offline bullying) in the future to capture a more complete picture of the situation.